# Current Advances of Nitric Oxide in Cancer and Anticancer Therapeutics

**DOI:** 10.3390/vaccines9020094

**Published:** 2021-01-27

**Authors:** Joel Mintz, Anastasia Vedenko, Omar Rosete, Khushi Shah, Gabriella Goldstein, Joshua M. Hare, Ranjith Ramasamy, Himanshu Arora

**Affiliations:** 1Dr. Kiran C. Patel College of Allopathic Medicine, Nova Southeastern University, Davie, FL 33328, USA; jm4719@mynsu.nova.edu; 2John P Hussman Institute for Human Genomics, Miller School of Medicine, University of Miami, Miami, FL 33136, USA; axv492@miami.edu (A.V.); JHare@med.miami.edu (J.M.H.); 3Department of Urology, Miller School of Medicine, University of Miami, Miami, FL 33136, USA; ojrosete@med.miami.edu; 4College of Arts and Sciences, University of Miami, Miami, FL 33146, USA; khs55@miami.edu; 5College of Health Professions and Sciences, University of Central Florida, Orlando, FL 32816, USA; gabigoldstein98@gmail.com; 6The Interdisciplinary Stem Cell Institute, Miller School of Medicine, University of Miami, Miami, FL 33136, USA; 7Department of Medicine, Cardiology Division, Miller School of Medicine, University of Miami, Miami, FL 33136, USA

**Keywords:** immunotherapy, nitric oxide, prostate cancer, castration, checkpoint inhibitors

## Abstract

Nitric oxide (NO) is a short-lived, ubiquitous signaling molecule that affects numerous critical functions in the body. There are markedly conflicting findings in the literature regarding the bimodal effects of NO in carcinogenesis and tumor progression, which has important consequences for treatment. Several preclinical and clinical studies have suggested that both pro- and antitumorigenic effects of NO depend on multiple aspects, including, but not limited to, tissue of generation, the level of production, the oxidative/reductive (redox) environment in which this radical is generated, the presence or absence of NO transduction elements, and the tumor microenvironment. Generally, there are four major categories of NO-based anticancer therapies: NO donors, phosphodiesterase inhibitors (PDE-i), soluble guanylyl cyclase (sGC) activators, and immunomodulators. Of these, NO donors are well studied, well characterized, and also the most promising. In this study, we review the current knowledge in this area, with an emphasis placed on the role of NO as an anticancer therapy and dysregulated molecular interactions during the evolution of cancer, highlighting the strategies that may aid in the targeting of cancer.

## 1. Introduction

Nitric oxide (NO) is a molecule with a very short half-life, produced by the action of nitric oxide synthases. Since NO was first discovered as being identical to endothelium-relaxing factor, the number of biochemical and physiological processes that undergo some form of NO signaling has continued to grow. When NO was observed to influence the development, growth, and metastasis of tumor cells, many studies emerged that were in direct conflict with one another. For many years, debate raged within the community about whether NO was tumoricidal or carcinogenic. However, as the body of scientific literature grew, the role of nitric oxide within carcinogenesis has been more clearly defined. Unfortunately for those seeking to make therapeutics, nitric oxide appears to have the capability to be both tumor-promoting and tumoricidal. NO’s bimodal effects on different cancer types is a phenomenon best termed as the Yin and Yang of NO [1,2,3]. Determining which effect predominates is complex and often depends upon the tissue NO exerts its effects, the concentration of NO administered, and tumor microenvironment. Nevertheless, these discoveries have led to a wide number of proposed uses for NO as an anticancer agent, either alone or in combination with other treatment modalities [4]. Here, we seek to outline the complexities of NO signaling within carcinogenesis and tumor progression at the biochemical and physiological levels. Furthermore, we also discuss the impact of NO in cancer therapy and outline its role as an emerging anticancer agent.

## 2. Physiology of NO 

### 2.1. Chemical Properties of NO

Nitric oxide is a diatomic free radical molecule with high reactivity across a wide range of biomolecules. High output of nitric oxide leads to effects such as nitration, nitrosation, and oxidation, which can then affect cellular functioning. Nitric oxide can interact with oxygen or oxide ions to form reactive nitrogen species such as dinitrogen trioxide and peroxynitrite [5]. It can also react with nitrogen dioxide to form dinitrogen trioxide and can react with superoxide to form peroxynitrite [6]. Both molecules can cause DNA damage through nitrosative and oxidative stress. Both molecules can cause DNA damage through nitrosative and oxidative stress. Dinitrogen trioxide can lead to formation of N-nitrosamines through nitrosation of amines and then alkylate DNA. This alkylation of primary amines can lead to the formation of diazonium ions and further deamination [7]. Peroxynitrites can oxidize and nitrate DNA and also potentially cause single-strand DNA breaks due to an attack on the sugar-phosphate backbone [7]. The biological effects of nitric oxide are dependent upon myriad factors such as the formation of the molecule, its metabolism, types of nitric oxide synthases present, and concentration of nitric oxide present.

### 2.2. Synthesis of NO

The primary means by which cells synthesize NO (Figure 1) is through the conversion of l-arginine to l-citrulline via the enzymatic action of nitric oxide synthases (NOS) [8]. There are three isoforms of the NOS family—inducible NOS (iNOS/NOS2), endothelial NOS (eNOS/NOS3), and neuronal NOS (nNOS/NOS1). The eNOS and nNOS isoforms are constitutively expressed in a variety of cell types and can be activated as a result of calmodulin-binding due to a rise in intracellular calcium. The constitutive NOS isoforms become activated or inhibited by phosphorylation from protein kinases. Unlike the constitutive NOS isoforms, iNOS displays a higher affinity for calmodulin, and therefore its activation is not calcium-dependent. Among the three isoforms, iNOS, along with being calcium-independent, also produces high concentrations of NO in a shorter time frame [8].

Another more controversial mechanism of NO formation is through the nitrite-nitrate-nitric oxide pathway. Nitrates and nitrites are physiologically recycled in the blood and tissues to produce NO and other bioactive nitrogen oxides [9,10,11,12]. In the body, nitrate, primarily from diet and oxidation of NOS-derived NO, is actively taken up by the salivary glands and reduced to nitrite anion by commensal bacteria in the mouth [13,14]. Nitrite is further metabolized in blood and tissues into a variety of bioactive nitrogen oxides. This reduction is catalyzed through numerous pathways involving myoglobin/hemoglobin, ascorbate, xanthine oxireductase, and polyphenols [12,15,16,17,18,19,20,21,22]. The production of NO from these pathways is enhanced by hypoxic and acidotic conditions [23,24,25]. However, not all studies support the hypothesis that nitrite-nitrate derived nitric oxide contributes meaningfully as a biologically relevant signaling mode for downstream NO effects.

### 2.3. NO-Mediated Post-Translational Modifications

The dominant mode of NO signaling specificity occurs through post-translational modification of specific proteins, which regulates function in a manner analogous to post-translational phosphorylation [26]. S-nitrosylation is achieved through the covalent attachment of NO to the thiol side of a cysteine residue [27].

Proteins from almost all functional classes are substrates for S-nitrosylation. However, protein S-nitrosylation shows spatiotemporal specificity for certain cysteine residues within a protein [28]. While only a few cysteine residues are targeted at physiological amounts of NO, the modifications are generally enough to change the protein function, activity or interaction specificity [29]. For example, S-nitrosylation of cardiac ion channels modifies the channel’s dynamics and activity profile [30]. S-nitrosylation is a dynamic bidirectional process, and the relative balance of nitrosylated and de-nitrosylated proteins can serve as a biofeedback mechanism for physiological functions [30,31]. In the heart and vasculature, protein de-nitrysolation via S-nitrosoglutathione reductase (GSNO-R) regulates vascular tone and β adrenergic activity [31], the processes of S-nitrosylation and de-nitrosylation are important regulatory mechanisms for carcinogenesis and metastasis and may be responsible for ischemic episodes and more [32,33,34,35]. Importantly, hemoglobin is a key target of S-nitrosylation, whereby S-nitrosylated thiols participate in the allosteric shift that regulates the process of oxygen loading and unloading [36]. S-nitrosylation is critical for NO to exert its pro-cancer effect in many malignancies, as the downstream consequences of S-nitrosylation can cause aberrant signaling, which over time may lead to unchecked growth, angiogenesis, and metastasis [37].

Another type of post-translational modification mediated by NO is S-glutathionylation. S-glutathionylation is a reversible process that involves the addition of a proximal donor of glutathione to thiolate anions of cysteines in the target proteins. This modification alters the mass, charge, structure, and/or function of the protein and may also prevent degradation via sulfhydryl overoxidation or proteolysis [38]. Presence of s-glutathionylated serum proteins can be used as a biomarker in individuals exposed to oxidative or nitrosative stress-causing agents [38]. This process is a candidate mechanism for controlling the generation of reactive oxygen and nitrogen species associated with stress signaling and functional responses [26,38].

A third type of post-translational modification is mediated by NO via protein tyrosine nitration, although this is not widely accepted. This modification is caused by NO-derived oxidants like peroxynitrite and involves the formation of an intermediate tyrosine radical [39]. Nitration of protein tyrosine residues happens when the hydrogen at the third position in the phenolic ring is substituted by a nitro group, forming 3-nitrotyrosine. The formation of this product shows an oxidative modification that favors pro-oxidant processes [40]. The two major mechanisms leading to tyrosine nitration in vivo are the formation of peroxynitrite and the production of nitrogen dioxide by heme proteins [41]. Overall, tyrosine nitration is proposed to cause profound structural and functional changes in proteins and might serve as a marker for nitrosative stress [42,43].

### 2.4. NO cGMP Signaling Pathway

The nitric oxide-cyclic GMP-protein kinase G (PKG) cascade is recognized as an endogenous apoptotic pathway in many cancer types [44]. Two distinct types of guanylate cyclases (GCs) catalyze the conversion of guanosine triphosphate (GTP) to cyclic guanosine monophosphate (cGMP). Particulate GCs are transmembrane receptors for natriuretic peptides, whereas cytosolic-soluble GC serves as a target receptor for NO [44,45]. NO, at nanomolecular levels, binds to prosthetic heme on the β subunit of soluble GC, forming a NO-GC complex [46]. NO-GC binding then increases conversion of GTP to cGMP. The increase in cGMP initiates downstream effectors including two serine-threonine kinases, PKG-I and PKG-II, which share common targets with protein kinase A (PKA) but activate PKA-independent pathways as well [47]. The NO-cGMP signaling pathway is the primary means by which NO serves as a vasodilator, as increasing cGMP concentration inhibits calcium influx into the cytoplasm, preventing smooth muscle contraction (Figure 2) [48].

### 2.5. NO and Redox Balance

The effect of NO is often balanced against reactive oxygen species (ROS), whereby localized changes in the concentration often inversely affect the other, termed nitroso–redox balance, which has been suggested to play a major role in heart failure [35,49]. Thus, elevations in superoxide, a ROS, lead to decreases in NO and vice versa. Low basal levels of NO are important for maintaining the redox state of the tissues, whereby reactions of NO and superoxide produce substrates that bind to S-nitrosylation sites on proteins [35,36]. Furthermore, at physiological concentrations NO also performs a host of other antioxidant functions, which can help prevent aberrant signaling [50]. However, pathophysiological processes upregulate NOSs, resulting in excess NO production, which can lead to abnormal signaling [51]. When NO production becomes elevated, it can cause nitrosative stress. Nitrosative stress, acting similarly in a manner to oxidative stress, can then affect homeostasis and alter protein function [35,50]. A full review of the mechanisms that underlie this process and how it contributes to human disease was previously published by Hare and Stamler [35].

## 3. NO and the Immune Response

### 3.1. NO and the Antimicrobial Response

NO was first discovered to play a role in immunological function as a byproduct of macrophage activation [52]. Traditional experiments focused on the role of iNOS, which was found to be upregulated in the alveolar macrophages of patients with tuberculosis and other infections [53]. Multiple studies reported that NO can contribute to host antimicrobial resistance, such as iNOS-deficient mice often, but not always, display exacerbated infections, and NO is upregulated in response to markers of infection such as lipopolysaccharide [53,54,55]. One proposed mechanism for NO’s antimicrobial effect is the creation of powerful free radicals. These free radicals, such as peroxynitrite, can directly damage pathogens through phagosomal redox chemistry, although they can also affect S-nitrosylated pathogen proteins, modifying critical components required for proper cellular functioning [56,57]. Additionally, NO is found to bind to a wide array of intracellular molecules, including transcription factors and inorganic molecules, which may aid in infection clearance [52]. However, inhibition of iNOS may actually aid in recovery from certain infections, and NO was implicated in influenza-mediated pulmonary injury [53,58]. Evidence also suggests that macrophages manage their own internal concentration of NO and that too much NO can have potentially harmful intracellular effects [57,59,60]. These self-protective mechanisms may be enhanced by iNOS compartmentalization to phagosomes, preventing iNOS from acting on other compartments [61]. These results provide evidence that NO mediates immune responses, although NO’s exact effect can be variable and is tightly regulated intracellularly.

### 3.2. Pro-Inflammatory Response

NO’s function in inflammation is complex, and NO has been implicated in both immunosuppressive and immunopathological effects [52]. iNOS is often upregulated in response to inflammation, as one study of sepsis suggested that NO levels were elevated in response to higher concentrations of pro-inflammatory cytokines, such as TNF-α, which in turn could directly upregulate iNOS through nuclear factor kappa B (NF-κB) signaling [62,63]. Additionally, anti-inflammatory cytokines, such as IL-10, were associated with lower NO levels [62]. NO is consistently elevated during chronic inflammatory states, which can lead to cellular apoptosis through some combination of cytochrome *c* release, p53 activation, or Bcl-2 associated X /Bcl-2 homologous antagonist killer protein (BAX/BAK) recruitment [64]. However, if NO levels become inordinate, NO can also lead to necrosis [64]. The difference between NO leading to apoptosis or necrosis is hypothesized to be due to a peroxynitrite concentration-dependent mechanism, with overwhelming levels of peroxynitrite inevitably leading to necrosis [64]. The discovery of NO’s central role in inflammation led to the hypothesis that NO and iNOS may contribute to the pathogenesis of many different diseases, including those in the heart, pulmonary system, vascular dynamics, and the blood [35]. Indeed, elevated concentrations of NO and inflammatory cytokines are found in a plethora of conditions, which provides evidence that NO may be essential for some pathogenic effects, although this is controversial [64,65,66,67].

### 3.3. Anti-Inflammatory Effects

Such controversy has endured because NO is involved in immune autoregulation across many different cell types and intracellular processes. At the molecular level, NO has been shown to downregulate its own production through inhibition of NF-κB [68]. Furthermore, NO has also been shown to attenuate cytokine activating molecules such as IL-1β-converting enzyme and interferon-γ (IFN-γ)-inducing factor, suggesting that NO can simultaneously aid in the immune response and tune the immune response [69]. Interestingly, T lymphocytes do not express iNOS directly, but inhibition of iNOS directly affects their functioning [70]. NO downregulates T-helper cell 1 (Th1) response through selective inhibition of IFN-γ, IL-2, P-selection, and the cellular adhesion molecules intracellular adhesion molecule-1 (ICAM-1) and vascular cell adhesion molecule (VCAM) [71]. NO also increases the production of IL-4, prostaglandin-E2, and IL-12, which selectively support T-helper cell 2 (Th2) production [71]. When combined with prior work which found that NO can selectively encourage immune cell apoptosis and downregulate antigen presentation, these findings strongly implicate NO as a key mediator of autoimmune responses [52,71,72]. However, these findings remain controversial as many labs have found null or opposite results [72]. Still, a number of animal models suggest that iNOS inhibition is directly correlated with decreased suppression of autoimmune diseases and some exhibited increased severity, providing system-level evidence of NO’s immunoregulatory function [71].

## 4. NO and Carcinogenesis

### 4.1. Carcinogenesis Overview

Similar to its other immune functions, NO has been shown to simultaneously be tumoricidal and tumor-promoting (Figure 2) [73]. NO is highly reactive with nucleic acids and can induce mutations upon chronic exposure [74]. One hypothesis suggests that NO-induced mutations from iNOS upregulation may drive carcinogenesis from chronic inflammatory processes [74]. However, these mutations may then induce p53 mediated growth arrest, DNA repair enzyme activation, and iNOS downregulation [7]. Therefore, p53 mutations may be a critical inflection point that steers NO away from growth arrest and apoptosis towards carcinogenesis [74]. Importantly, cell types inside a tumor may not be homogenous, and they contain a vast network of connective tissue, blood vessels, and immune cells, termed the tumor microenvironment (TME), which may also be differentially regulated by NO [74,75]. Furthermore, NO may modify tumor cell metabolism inside the microenvironment by promoting the Warburg effect and chemotherapeutic resistance [76]. Upon tumor formation, NO promotes angiogenesis, downregulates immune surveillance and encourages metastasis with gross pathological specimens demonstrating markedly elevated levels of NO or iNOS in a wide variety of malignant tumors. Increased iNOS expression has been linked to poor patient outcomes for multiple types of malignancies and thus has become an important target for future therapeutic delivery [73].

### 4.2. NO Biochemistry within the TME

NO is an important immune modulator of the TME. Innate immune cells such as macrophages and NK cells upregulate production of NO during tumor cell invasion [77,78]. Within the TME, NO is produced mainly by iNOS, expressed in macrophages and tumor cells, and to a much lesser extent by eNOS and nNOS. NO is the main chemical macrophages utilize to mount an immune attack against tumor antigens, and functions to activate apoptosis [79]. NO can bind to the heme copper center of cytochrome c oxidase, thereby oxidizing the enzyme and inhibiting the mitochondrial respiratory chain, leading to permeability and escape of cytochrome c into the cytoplasm [80,81]. Additional mechanisms of apoptosis activation include phosphorylation of p53, a tumor suppressor antigen, p53 retention in the nucleus, and s-nitrosylation of NF-kB, an anti-apoptotic factor, inhibiting its ability to bind to the DNA [80,81]. However, NO might not only exert cytotoxic effects but is actually more dichotomous in nature with both tumor-promoting and tumor-suppressing effects. At high concentrations, NO is able to induce death, while at low concentrations it may actually protect cells from death. At 50–100 nM, NO is able to phosphorylate extracellular signal regulated kinase (ERK) activate the AKT pathway, and stabilize HIF1a, while at the 300 nM–1 μM range, NO induces DNA damage, p53 activation, and causes nitrosative stress [56].

### 4.3. NO and Angiogenesis in the TME

Angiogenesis occurs in hypoxic core regions of the tumor due to lack of oxygen and nutrients, which may be intricately regulated by the presence of NO. Hypoxia-inducible factor 1 and 2 activate the expression of pro-angiogenic growth factors such as VEGF, FGF-2, IL-8, PDGF, IGF2, and TGFβ. HIF1α and HIF1β are usually degraded in normoxic conditions, but are stabilized in hypoxic conditions (<5% oxygen) as well as by low levels of NO and reactive nitrogen species (RNS) [82,83,84]. Low levels of NO may also regulate endothelial cell fate by activating GC and increasing cGMP levels (Figure 3), which acts directly on the endothelium to cause its reorganization into vessel-like formations [85]. Additionally, low levels of NO are able to activate matrix metalloproteinases (MMP-1, -9, -13), which degrade the components of the extracellular matrix and paves the way for tumor cell dissemination outside of the tumor and endothelial cells into the tumor core for blood vessel formation [86,87]. Thus, low NO is beneficial for ensuring upregulation of pro-angiogenic factors.

### 4.4. NO and Immune Cells within the TME

Macrophages can switch between different phenotypes and perform completely opposite functions in the TME based on their inflammatory potential. For instance, M1 macrophages are induced by pro-inflammatory cytokines (tumor necrosis factor-alpha TNFα, interferon-gamma INF-γ, Interleukin-1b IL-1β), activate Th1 immune response, and induce their cytotoxic effects by upregulation of iNOS that produces toxic concentrations of NO [77,78]. In fact, NOS2 expression is a hallmark of M1 macrophages. Alternatively, activated M2 macrophages are less immunologically inflamed and often act as housekeeping cells, which sweep apoptosis debris after pathogen clearance. These are activated by signals from anti-inflammatory cytokines such as tumor growth factor-beta (TGFβ) and anti-inflammatory interleukins (IL-10, IL-13), which downregulate iNOS production and reduce the activity of M1 macrophages and T helper cells [88,89]. Given their anti-tumor potential and ease of isolation, exogenous infusion of M1 macrophages has been investigated as a potential autologous therapy approach [90,91]. One study utilized ex vivo LPS-stimulated macrophages to autologously re-infuse into patients; however, results showed no significant clinical improvement with minor side-effects [92,93]. Another study using INF-γ-stimulated macrophages showed reduced metastasis and tumor growth but no significant tumor regression in mouse studies [93,94] and subsequent human clinical trials [95]. These conflicting results may need to account for the presence of anti-inflammatory mediators (TGFβ, IL-10, PGE2) and the hypoxic microenvironment that inhibits iNOS expression, which neutralize M1 infusions or skew them back to the immunologically suppressive M2 type [96]. Thus, finding ways to increase production of NO by tumor cells and macrophages, thereby activating immunity toward immunologically active M1 macrophages, might be necessary to effectively eradicate the tumor (Figure 4).

Aside from their NO-producing ability, macrophages are among the most prominent immune cells in the TME, as they are able to infiltrate deep into the tumor and can account for almost 50% of the tumor mass [91,97]. Tumor-associated macrophages (TAMs) reside in the most hypoxic and necrotic regions of the tumor, where they are known to assist in tumor progression [98]. TAMs express pro-angiogenic growth factors and matrix metalloproteinases MMPs (VEGF, PDGF, FGF-2, MMP-7, MMP-12), as well as create an environment that makes M1 macrophages and CD4+ and CD8+ T cells unresponsive to tumor antigens. Another type of macrophage, Tie2-expressing monocytes (TEMs), resides close to blood vessels, where they function in a similar way to TAMs by promoting angiogenesis via upregulation of VEGF and MMPs [99]. Depletion of TEMs results in marked inhibition of angiogenesis, so these are required for metastasis [100]. Thus, tumor-infiltrating macrophages are more similar to the anti-inflammatory M2 type, which suggests that the majority of macrophages in established tumors are anti-inflammatory and thrive in immunologically suppressed conditions.

Chronic inflammation has been linked to multiple forms of malignant transformation, and NO produced by different immune cells during acute infections plays a significant role in this process. Since it is extremely small and lipophilic, NO is able to quickly diffuse through cell membranes, oxidize DNA, deaminate bases, and deactivate proteins in the DNA repair machinery through s-nitrosylation. Additionally, long-term this genomic instability might activate oncogenes and inhibit tumor suppressor genes [101,102]. In a p53 knockout mouse model, increased NO production due to iNOS upregulation through a negative-feedback loop accelerated spontaneous tumor formation [103]. In addition, high amounts of NO in a breast cancer line activated expression of c-Myc, a potent oncogene [104]. Furthermore, deactivation of retinoblastoma (Rb) tumor suppressor gene by NO-driven hyperphosphorylation was shown in a mouse model of colitis [105]. Thus, prolonged episodes of acute infection can lead to overall genomic instability and activation of undesirable oncogenes as well as inhibition of tumor suppressor genes.

In the immunogenically inactive tumors, usually occurring during escape of tumor cells and metastasis phase, NO production is lowered to ensure decreased immunogenicity. This occurs by several processes. Anti-inflammatory cytokines in the TME actively lower the expression of iNOS and degradation of iNOS mRNA by hypoxic conditions within the tumor core [106]. Moreover, NO produced by TAMs is often captured by circulating erythrocytes, where it oxidizes hemoglobin iron centers and thus reduces circulating NO even further [107]. Regardless of which mechanism is responsible, immunologically suppressed tumors show reduced expression of NO. These low amounts of NO usually modulate apoptosis protein cascades through s-nitrosylation of caspase-3 and Bcl-2 [108,109]. Thus maintaining low/transient levels of NO and ensuring that apoptotic pathways are effectively inhibited is one potential mechanism by which tumors are able to increase their proliferation capacity.

### 4.5. NO as a Biomarker in Carcinogenesis

Estimation of iNOS expression in solid tumors has attracted interest for biomarker use; however, there are multiple levels of iNOS regulation that must first be considered. Different iNOS isoforms are under transcriptional regulation due to multiple transcription binding sites within the promoter of the gene, and these sites are susceptible to regulation by various cytokine mediators [110,111]. iNOS promoters are also different among species; in the mouse, induction with lipopolysaccharide (LPS) and pro-inflammatory cytokines (TNFα, INFγ, IL-1b) is enough to upregulate iNOS expression, while a more complex scheme is needed to achieve the same effect in humans [112]. The stability of iNOS mRNA is also under tight regulation by various cytokines that upregulate RNA binding proteins which compete for binding to the 3′-UTR region, thus stabilizing or destabilizing mRNA transcript [111]. Additionally, iNOS transcripts might be targeted through cytokine signals (SOCS-1) by small non-coding RNA molecules (miRNA) such as miR-155 [113] and miR-146 [94,114], resulting in translational inhibition. Finally, ready availability of L-arginine as a substrate for NO production is crucial, as it is mostly supplemented through the diet, and the bioavailability of necessary co-factors for proper enzymatic activity such as tetrahydrobiopterin (BH4) may also influence iNOS activity [111]. Thus, the use of iNOS expression as a prognostic biomarker, though promising, is currently controversial.

What complicates the issue further is that iNOS expression might not be as well correlated with cancer progression as previously thought and finding methods that show an accurate estimation of NO production is currently a difficult task. Expression of iNOS can be measured using Western blot analysis, qPCR, or immunohistochemistry, though iNOS mRNA is subject to degradation in paraffin-embedded blocks. Moreover, iNOS expression is not always indicative of appreciable NO production, depending on the tumor niche and cytokine content, as already mentioned. Thus, measuring the activity of the enzyme might be more beneficial; however, this is limited to the availability of fresh tissue. Different techniques have been developed, such as estimation of nitrates and nitrites [115], measuring conversion of radiolabeled (H^3^) L-arginine to (H^3^)-L-citrulline [116,117], or immunohistochemical detection of nitrotyrosylated proteins [118]. Needless to say, these techniques are tedious and expensive to implement, and might not be a direct measure of NO production. Several studies have shown a correlation between iNOS expression and tumor stage progression in malignant melanoma [119,120], poor survival in colorectal cancer [118,121], poor prognosis in estrogen receptor (ER)-negative breast cancer patients [122,123], and lymph node metastasis in pancreatic cancer [124]. However, in large hypoxic tumors, NO production was shown to be reduced when compared to smaller immunologically active tumors [94]; thus, iNOS expression by overall tumor mass might indicate a gradient of concentrations, which further hinders its prognostic potential [115]. Interestingly, in the lungs, excess exhaled NO from iNOS has been proposed as a biomarker of lung cancer, which was significantly higher in those with lung cancer [125,126,127]. However, exhaled NO is notably upregulated in other inflammatory airway conditions, particularly in asthma and heart failure, and its specificity and sensitivity for lung cancer remain under investigation [126,128,129]. Therefore, iNOS biomarker potential, though attractive, necessitates the development of better tools and a clearer understanding of its expression patterns within solid tumors.

## 5. NO in Different Cancer Types

### 5.1. Lung Cancer

Lung cancer has the second-highest incidence amongst all cancer types, with a poor 5-year survival rate of 4–17% [130,131]. Due to its high incidence and often aggressive course, it is no surprise that NO has been shown to function in the pathogenesis of lung cancer. In fact, population-based studies of NO and NO metabolites suggest that increased accumulation of NO metabolites were associated with an increased risk of lung cancer, even after controlling for relevant confounding factors such as smoking [132]. Furthermore, cigarette smoking, a notable risk factor for lung cancer, often contains NO and other ROS [133]. Such compounds may encourage the development of NO-mediated cellular changes, which can eventually accumulate in malignant lung tissue (Figure 5) [133]. Excess exposure of lung tissue to NO results in the accumulation of nitrosylated proteins, which were significantly higher in those with lung cancer [134]. Often, this effect is mediated via iNOS, which is often shown to be upregulated in lung cancer cell lines, similar to other cancers [125,126].

Excess NO in lung tissues may contribute to a number of effects that synergistically act to encourage cancer formation. For example, one study noted that gaseous nitric oxide and inducible NOS produced an increase of 8-nitroguanine, which may increase DNA damage and encourage mutagenesis [135]. Although protein nitrosylation is often viewed as a marker of oxidative stress, one group proposed that protein nitrosylation may also impair antioxidant proteins and those involved in cellular metabolism, which may further contribute to the development of non-small-cell lung cancer (NSCLC) [136]. Furthermore, elevated iNOS in NSCLC cells are linked to p53 mutations, removing growth checkpoint inhibition and creating cells with unlimited replicative potential, another notable effect of NO [137,138]. Within lung cancer, the antiapoptotic effect is hypothesized function through BCL-2 upregulation and modifications to FAS death-ligand signaling, which encourages aggressive growth [139]. Once established, NO also modulates integrin expression amongst NSCLC cells via protein kinase B (AKT) activity, and increased AKT activity is correlated with a poor prognosis and chemotherapeutic resistance [140]. Increased exposure to nitric oxide over time is itself correlated with enhanced cellular migration and chemotherapy resistance [141,142]. Nevertheless, a number of NO-donating compounds have been reported to have efficacy at inhibiting some of these effects [143,144,145]. As such, NO-donating drugs may raise the concentration of NO to cytotoxic levels, bypassing the physiological mechanisms underlying tumorigenesis and actually serve to inhibit cancer growth.

### 5.2. Breast Cancer

Amongst all non-skin cancer sites in females, breast cancer has the highest incidence and second-highest mortality, although the 5-year survival rate is over 90% [129]. Similar to NSCLC and other types of cancer, elevated levels of iNOS were noted from tissue samples of patients with breast cancer, which was significantly greater than normal breast tissue and benign breast diseases such as fibroadenomas [123]. NO was also shown to function through the same p53 and AKT pathways as those of lung cancer cells, suggesting common pathways for NO signaling to induce cancer growth and encourage progression [146]. Such pathways may work synergistically with other NO pathways in breast cancer, such as the epidermal growth factor receptor (EGFR)-mediated activation of ERK [147]. Furthermore, a number of studies suggest that the activity of NO pathways in breast cancer are concentration-dependent, which is backed by other NO studies. NO concentration of less than 100 nM activates cGMP-dependent pathways, while concentrations of 200–600 nM activate cGMP independent pathways [146]. Above these concentrations, NO has been suggested to phosphorylate p53 and halt the function of DNA repair enzymes [146]. cGMP dependent pathways were noted to be dysregulated within breast cancer, with lower concentration of cGMP being correlated to malignant disease, suggesting low physiological concentration of NO-mediated cGMP signaling may actually provide protection against breast cancer [148]. cGMP independent pathways have been discussed previously and are not unique to breast cancer. Nevertheless, NO-mediated HIF-1a stabilization, nitrosylation of metabolic enzymes, and induction of VEGF are major pathways of breast cancer growth and metastasis [146]. Additionally, NO-mediated activation of matrix metalloproteinases (MMP) and inhibition of tissue inhibitor of matrix metalloproteinases-1 (TIMP-1) may reorganize the tumor microenvironment to increase metastatic potential, while further stabilizing downstream cGMP-independent NO pathways [146]. Numerous studies have found that many of these NO-mediated effectors, such as VEGF and CXCR4, are correlated to lymph node metastasis and overall prognosis [149,150]. The growth arrest and p53 phosphorylation exhibited by supraphysiological concentrations of NO may explain the effect that NO-conjugated drugs have at inhibiting breast cancer growth and inducing apoptosis [151]. The underlying mechanisms behind how nitric oxide becomes upregulated in breast cancer remain unknown; however, similar to other cancer types, systemic effects from environmental, hereditary, and physiological processes all may play a role. For example, iNOS was noted to be upregulated in response to glucocorticoids, a stress hormone, suggesting multisystem processes may contribute to the pathogenesis of breast cancer [152]. Furthermore, cyclooxygenase-2 (COX-2) and NO are strongly linked in breast cancer, with one multivariate analysis suggesting that those with COX-2 and iNOS-positive tumors were associated with extremely poor prognosis [153]. Nevertheless, the exact role of NO in breast cancer remains controversial, especially in the context of estrogen- and progesterone-mediated signaling.

Estrogen and progesterone have also been shown to modulate nitric oxide expression, through the actions of eNOS and iNOS [154,155]. Estrogen is thought to downregulate the expression of iNOS and upregulate the expression of eNOS, which in turn activates phosphoinositide 3-kinase (PI3K) and AKT through direct activation and increased transcription, which have been shown to be critical for cancer proliferation [156]. Furthermore, upregulation of eNOS appears to be a unique feature of estrogen-dependent tumors, which are lacking in ER- breast cancers [157]. Interestingly, progesterone’s function within nitric oxide synthesis appears to be more controversial, with several studies suggesting that progesterone can affect eNOS production and others suggesting null or contradictory effects [156,158,159]. For example, one study suggested that progesterone-induced iNOS expression in vitro, which then induced cell apoptosis [160]. Additionally, progesterone has been proposed to affect and downregulate estrogen-mediated nitric oxide production [156]. These results suggest that breast cancer tumor progesterone status is a favorable prognostic factor, an effect that is backed by clinical studies [161]. Although it is currently unknown exactly how the surface marker human epidermal growth factor receptor 2 (HER2/neu) affects nitric oxide production, one group suggests that HER2/neu downregulates nitric oxide production, ablating the apoptotic effect of chemotherapeutics in vitro, potentially through a COX-2 mechanism [162,163]. Nevertheless, triple-negative breast cancer, the most aggressive form, appears to also be strongly correlated to iNOS [164]. Thus, nitric oxide expression appears to be central to the pathogenesis of breast cancer, regardless of receptor phenotype expression.

### 5.3. Prostate Cancer

Prostate cancer is highly prevalent in men and has an age-adjusted incidence of 453.8 per 100,000 and is highest amongst those 65–74 [165]. Similar to other cancers, iNOS expression was also significantly increased in prostate adenocarcinoma when compared to healthy prostate tissue [166]. Furthermore, previous work demonstrated that iNOS expression was greatest amongst those with metastasis and high Gleason scores, and one meta-analysis found that tumor iNOS expression may serve prognostic value [167,168]. These findings are backed by clinical studies, which suggest greater nitrosative stress in those with prostate cancer as compared to benign prostatic hyperplasia (BPH) and controls [169]. However, eNOS also appears to be linked to prostate cancer, as some but not all studies have shown that genetic polymorphisms of iNOS and eNOS carry an increased risk of high Gleason score prostate cancer [170,171,172]. eNOS may in turn upregulate pleiotrophin (PTN), expression through ERK activity, increasing tumor and endothelial migration, laying the groundwork for metastatic disease [173]. Additionally, the eNOS complex can interact with the estrogen receptor to transcriptionally alter other important gene products, such as Glutathione transferase P1, which are correlated to disease pathogenesis [174].

Prostate cancer is often initially androgen-sensitive, which is useful in androgen deprivation therapy, and is often a first-line therapy for advanced and metastatic disease [175]. This advanced form of prostate cancer is termed castration-resistant (CRPC) and is associated with poor outcomes [176]. Previous studies have implicated NO as a potential mediator of androgen resistance by androgen receptor transcriptional suppression and direct androgen receptor inhibition, which were mediated by iNOS and eNOS, respectively [177,178]. iNOS induction may encourage tumor growth, as two studies from the same lab found that NO promotes survival and accelerates tumor growth after oxidative stress [179,180]. However, testosterone, an androgen, also increases NO concentration and survival of prostate cancer cells, suggesting that NO’s mechanism in androgen sensitivity and resistance remains elusive [1,2,181,182]. These androgen-specific mechanisms may interact with previously defined effects of NO on hypoxia-induced HIF-1a and tumor angiogenesis, compounding the effect of nitric oxide on prostate cancer [183].

### 5.4. Gastrointestinal Cancers

When summed, digestive system cancers have the highest incidence amongst all organ systems with colorectal cancer having the third-highest incidence amongst all non-skin types, regardless of gender [129]. Interestingly, one analysis of gastrointestinal cancers suggested that most had adherent iNOS expression, although whether iNOS expression was upregulated or downregulated was dependent upon cancer type [184]. Similar to other cancers, gastrointestinal cancers may also use previously discussed NO signaling pathways such as PI3-K, p53, AKT, PTEN, NF-kB, MMPs, and HIF-1a in their carcinogenic pathogenesis [184]. Malignant transformation of colorectal and other cancers are often dependent upon the epithelial–mesenchymal transition (EMT) in which cancer cells express genes that are normally associated with connective tissue [185]. Although a number of cell signals can stimulate this pathway, one particularly important player is the APC/Wnt/β-catenin pathway, which is often dysregulated in colorectal cancer [186]. In normal healthy colonic tissue, APC downregulates B-catenin, which prevents polyp formation [186]. NO has also been shown to upregulate the Wnt/β-catenin pathways, potentially through negative feedback NF-κB response elements on a Dickkopf-1 gene promoter that reduces gene silencing [187]. iNOS has been previously suggested to influence the function of NF-κB, and these findings correlate to the increased iNOS expression, which is often noted in colorectal carcinomas. However, increased iNOS expression is not a ubiquitous finding across all studies [188]. In addition, APC and the wingless-related integration site (WNT)/B-catenin pathways also serve to regulate COX-2 in a similar manner to NO, suggesting that NO may work in synchrony with COX-2 to promote its pro-cancer effects [184,189]. COX-2 has been previously linked to many of the same pro-cancer effects as NO [190]. Nevertheless, the underlying mechanisms linking COX-2 to NO remain elusive, although both undergo NF-kB regulation [191]. Taken together, these findings suggest that regulation of NO and APC may occur simultaneously, and treatment strategies targeting these pathways may prevent WNT/B-catenin upregulation, preventing the development of cancer.

Similar to cigarette smoking, the processing of digested metabolites and chronic infectious agents may also lead to cancer through nitric oxide-mediated mechanisms. For example, one of the major risk factors for gastric cancer is dietary consumption of nitro-compounds, with a relative risk of 1.31 (95% confidence interval (CI), 1.13–1.52) for nitrites, and 1.34 (95% CI, 1.02–1.76) for NDMA, although nitrate consumption was associated with a lower risk of gastric cancer 0.80 (95% CI, 0.69–0.93) [192]. Dietary consumption of salivary nitrites exposes nitrites to stomach acid and ascorbic acid, which then produces nitric oxide, which can diffuse rapidly to surrounding tissue [193]. A similar increase in NO production was noted with chronic *Helicobacte pylori* infection, suggesting that enhanced nitric oxide exposure from environmental and infectious agents may be responsible for the development of cancer, especially when there is high or chronic levels of exposure to these agents [194]. Viruses were also not immune to NO, as Hepatitis B was shown to be inhibited by NO via INF-γ, and chronic infection with Hepatitis B may induce increased NO, which can predispose hepatocytes to mutagenesis, potentially through a c-jun, an n-terminal protein kinase (JNK) [195,196]. Furthermore, Hepatitis C was also shown to induce DNA damage through upregulation of iNOS and nitrosylation of DNA glycosylase [196,197]. The effects of chronic viral-induced upregulation of NO may be compounded by mutations to proteins regulating nitrosylation, like GSNOR, which can cause the buildup of formaldehyde, a well-known carcinogen [196], increasing the risk of liver cancer. Evidence also suggests that iNOS and NO may be dysregulated upon alcohol exposure, as iNOS and NO are upregulated in response to ingested toxins [196,198]. When combined with studies on NO, such results suggest that increased nitric oxide dysregulation in response to cigarettes and alcohol may be one mechanism for the increased cancer risk amongst patients who engage in these behaviors.

### 5.5. Other Cancers

NO dysregulation is linked to a wide range of malignancies, including those of the brain [199], genitourinary system [200,201], skin [202], thyroid [203], and head and neck [204,205] cancers. Due to the complexity of nitric oxide and its wide number of potential interactions, it is suggested that the multifactorial effects of NO are cancer-specific. Although NO may often work through a common pathway such as p53 to induce mutagenesis, the exact mechanisms often differ between cancer types. Furthermore, in cases of sporadic cancers with minimal risk factors and no underlying conditions, it is unknown exactly what causes the underlying upregulation of NO or if the elevated NO is in response to another effect. However, chronic unchecked NO signaling is clearly beneficial to tumors and can encourage progression and metastasis. Therefore, therapies that can control NO growth signals have great promise, although delivering such therapies to the tumor without affecting nearby or distant healthy cells remains a significant problem.

## 6. NO in Anticancer Therapy

### 6.1. NO Donors

NO donors function by increasing NO or NO isoforms (NO- or NO+) without the need for endogenous production [206]. NO donors work through a number of different mechanisms; however, the end result is often still the same, i.e., an increase in NO concentration within tissue beds [207,208]. Such modulation of NO concentration has lead to a number of proposed clinical uses for NO donors outside of cancer (Table 1) [209,210,211,212,213,214,215,216,217,218,219,220,221,222]. The blood vessels are particularly sensitive to the effect of NO, as the systemic veno- and vasodilator effects are useful for treating angina, acute coronary syndromes, and other cardiovascular diseases [223]. In cancer therapy, NO donors are particularly helpful as chemo- and radiotherapeutic sensitizing agents [224]. Advanced malignancies are often characterized by incomplete vascularization, which induces localized hypoxia. The resulting hypoxia stimulates the hypoxia-inducible factor1α (HIF-1α) pathway, priming cancer cells for survival against a variety of cellular death mechanisms induced by radio- or chemotherapy including autophagy, apoptosis, and DNA damage [224,225]. NO donors attempt to reverse this effect by increasing tumor perfusion, enhancing the effect of antitumor therapy [224].

Multiple clinical studies (Table 2) have demonstrated the efficacy of nitroglycerin (GTN), an organic nitrate, within the treatment of cancer. For example, in one phase II clinical trial, transdermal GTN improved outcomes for patients with advanced non-squamous cell lung cancer [226]. Not only has GTN shown efficacy in treating NSCLC, but GTN has demonstrated clinical effectiveness in liver, colorectal, and prostate cancer as well [227,228,229,230,231,232,233]. Furthermore, other preparations utilizing isosorbide mononitrate have also been attempted, although this approach did not show efficicacy in two clinical studies [234,235]. Although arginine, a nitric oxide precursor, is not traditionally viewed as a nitric oxide donor, a number of clinical trials have been completed examining the effectiveness of nutritional arginine for cancer (NCT04564521, NCT00559156, NCT02655081, NCT02844387). However, none currently have shown results.

HIF-1α-mediated resistance has been demonstrated to be vital for tumor resistance to a number of chemotherapeutics, although a number of specific mechanisms have been described that are often tumor-type-dependent [225]. NO donors may be similarly useful in the treatment of other cancers by targeting the underlying HIF-1α-mediated chemotherapy resistance, although no active clinical trials that test the effectiveness of adding a NO donor alone to chemotherapy are underway [225]. Importantly, the exact effectiveness of NO donors may vary by the type of cancer being treated and the NO donor being administered, making the ideal therapeutic to use NO donors in cancer therapy elusive. Nevertheless, an incredible number of preclinical in vitro and in vivo studies have demonstrated considerable efficacy in using a wide range of NO donors to treat cancer (Table 3). Amongst all NO donors that have not undergone clinical studies, diazeniumdiolates have shown the most promise. For example, DETA/NO demonstrated reversal of chemotherapeutic resistance for 5-fluorouracil (5-FU), doxorubicin, cisplatin, and fludarabine, and sensitized prostate cancer cells to TRAIL-mediated apoptosis and, when combined with a farnesyltransferase inhibitor, led to selective apoptosis in breast cancer cells [224,236,237,238]. Furthermore, the combination of PROLI/NO with carboplatin led to improved prognosis amongst rats with 6-c gliomas, which was thought to be due to improved chemotherapeutic delivery through the blood–brain barrier [239]. Preclinical studies also suggest that NO donors do not always have to be adjuvant to chemotherapy to demonstrate anticancer properties. One diazeniumdiolate, DETANONOate, was shown to inhibit the mesenchymal-to-epithelial transition and reverse the metastatic properties of the tumor [240]. A synthetic NO metal donor, ruthenium nitrosyl complex trans-[Ru(NO)(NH3)4(py)](PF6)3](pyNO), demonstrated considerable mitochondrial inhibition and an increase of ROS within the tumor, encouraging caspase-mediated cell death in liver cancer cells [239]. Arora et al. found that treating CRPC with GSNO, an S-nitrosothiol, reduced the expression of M2 macrophage expression within the TME, an important component of tumor progression [182]. Furthermore, other markers of cancer progression and resistance were suppressed, namely VEGF, the androgen receptor, and Androgen Receptor Splice Variant 7, while cytotoxicity increased through a greater number of M1 cytotoxic macrophages [182]. However, it should be noted that the potential mechanisms of NO donors as an anticancer agent are numerous (Table 3). Such studies suggest that NO donors have a number of novel mechanisms that may be beneficial in cancer therapy. However, NO often can affect multiple tissue beds simultaneously, including those outside of the tumor, and so consistent delivery of cytotoxic NO presents a major obstacle for researchers [207].

### 6.2. Phosphodiesterase-Inhibitors

Phosphodiesterases (PDEs) function as metallohydralses that catalyze the breakdown of cyclic adenosine monophosphate (cAMP) or cGMP into their inactive forms, 5′-AMP or GMP [264]. PDEs are thought to be involved with cancer progression and tumor growth because of the positive association they have with increasing tumor grade and stage as well as the decrease of cAMP and cGMP noted in many tumors [265]. Elevated PDE-5 levels have been documented in various types of human carcinomas including prostate, pancreatic, lung, colon adenocarcinoma, breast, and bladder squamous carcinoma [266]. PDE-5 inhibitors (PDE-5i) blunt the function of this critical recycling enzyme and block the breakdown of cGMP in 5′-GMP, which enhances the NO/cGMP signaling pathway [44]. Thus, PDE-5is function similarly to NO donors by enhancing the effect of NO on tissue; however, they rely on endogenous sources to maintain their effect rather than the exogenous NO provided by an NO donor [267].

PDE-5is are commonly used clinically to treat erectile dysfunction (ED); however, a wide range of PDE-is have shown anticancer activity, with many tested in clinical trials (Table 4) [265]. The primary differences between PDEs and the corresponding inhibitors that determine their functional significance are their different tissue bed distributions in addition to different regulatory feedback mechanisms and affinities for cGMP, cAMP, or both [265]. For example, thymoquinone, a natural herb with PDE-1i activity, has shown efficacy at inhibiting the growth of acute lymphoblastic lymphoma, cervical, and malignant central nervous system tumor cells, and an active clinical trial investigating the efficacy of thymoquinone to treat premalignant leukoplakia has recently been completed (CT03208790) [268,269,270,271,272]. Numerous other phosphodiesterase inhibitors (PDE-is) have also shown similar clinical effectiveness in preclinical studies [265]. Nevertheless, PDE-is may be limited by dose toxicity and systemic side effects unrelated to the primary tumor site. One large epidemiological study of 15,000 American men suggested that use of PDE-5is was associated with an increased incidence of melanoma; however, a retrospective meta-analysis found that although 4 out of 7 studies showed an increased risk of melanoma development with PDE-5i use, they failed to account for major confounders, and there was no linkage between the PDE-5i use and melanoma [273,274]. Interestingly, one report also suggests that PDE-5is can prevent the progression and development of prostate cancer, although other studies have shown null and even contradictory results [44,275,276,277]. Furthermore, the effectiveness of PDE-is as chemotherapeutics may be linked to their ability to enhance chemotherapy. Similarly to PROLI/NO, PDE-5is also demonstrated that they were able to increase the transport of doxorubicin across the blood–brain barrier in a rat brain tumor model, increasing the effect of chemotherapy [278]. In another study, the addition of sildenafil, a PDE-5i, with or without roflumilast, a PDE-4i, and theophylline, a methylxanthine, to lung cancer cell lines showed increased apoptosis and growth inhibition when given alongside a platinum chemotherapeutic [279]. Importantly, the same regime was not effective when combined with docetaxel, a taxane [279]. Taxanes function by disrupting microtubule disassembly, whereas platinum agents generate DNA double-stranded breaks through ROS generation. PDE-is may amplify NO’s downstream signaling cascade and increase the production of free radicals, which may then augment the free radicals produced from a platinum agent, thus increasing the efficacy of cancer. The potential anticancer mechanisms of PDE-is are numerous, and their anticancer properties were recently reviewed by Peng et al. [265]. Nevertheless, the addition of PDE-5i as adjuvant chemotherapy sensitizing agents may work through multiple mechanisms and is not strictly limited to one class of chemotherapeutic agents. Emerging evidence suggests that the sildenafil also improved the efficacy of docetaxel at treating CRPC, which the authors hypothesized was due to improved action of docetaxel on effector pathways, namely cGMP-mediated apoptosis and ERK/JNK downregulation [280]. Although numerous preclinical studies outline the potential roles for PDE-is in anticancer therapy, clinical trials (Table 4) have only begun to show evidence of efficacy [265,281]. Large, multicenter studies are needed before widespread clinical adoption of PDE-is in anticancer regimens.

### 6.3. Soluble Guanylate Cyclase Activators

Soluble guanylyl cyclase (sGC) is the receptor for NO, which binds to the ferrous (Fe2+) heme at histidine 105 of the β1 subunit [294]. Upon binding of NO, a remarkable increase in sGC activity is observed and cGMP production increases at least 200-fold [294]. In cancer, activation of sGC is notably impaired in a number of cancer cell lines including prostate, breast, and glioma, and restoration of sGC may decrease disease progression [294,295,296,297]. sGC activators, usually riociguat or vericiguat, stimulate the sGC irrespective of endogenous NO [298]. Although current clinical uses of sGC activators are limited to pulmonary hypertension and heart failure, other promising results exist for a host of other diseases including chronic kidney disease, systemic sclerosis, COPD, and even cancer [298,299,300,301,302,303,304,305].

Efficacy of sGC activators in cancer is not well studied, although two preclinical studies suggest that sGC can mitigate platinum chemotherapeutic resistance in oropharyngeal squamous cell carcinoma [304,305]. Nevertheless, some sGC activators were noted to be metabolites of pro-carcinogenic organic compounds, which the authors hypothesized could be partly responsible for carcinogenesis [306]. Furthermore, other studies suggested that peptides against sGC may actually be useful in treating androgen-independent prostate cancer [307]. Although of potential benefit for some cancers, the contradictory results of sGC activators provide further evidence of the dual-nature of NO in cancer pathogenesis. More preclinical studies are needed to demonstrate the efficacy or non-efficacy of sGC activator compounds in other cancers to make conclusions on their future potential as therapeutics.

### 6.4. Immunity Activators: PD-L1, PD-1, CSF1, and CSF1R

Immunotherapy is a developing area of cancer therapy with extremely promising results [308]. One such immunotherapy targets the programmed cell death-1 signaling cascade, at either the ligand (PD-L1) or the receptor (PD-1). PD-1 is commonly upregulated in response to immune activation and can be seen on the surface of CD4+, CD8+, and natural killer T cells, in addition to dendritic cells and B cells [309]. Meanwhile, the PD-L1 ligand is expressed on healthy peripheral tissues and serves to downregulate immunological response by decreasing proliferation, cytokine signaling, and overall survival of T-cells upon binding to PD-1 [309]. Lack of PD-1 or its ligand contributes to autoimmunity, as T-cells cells are free from peripheral immune surveillance feedback to continuously attack host tissues [309]. However, in cancer, the overexpression of PD-L1 allows the cancer to escape peripheral immune surveillance entirely, contributing to cell growth, immortality, and cancer progression [310]. Blockage of PD-L1 on cancer cells or PD-1 on immune cells via monoclonal antibodies has shown considerable benefit in clinical trials [311,312]. To date, three PD-L1 antibodies, atezolizumab, avelumab, and durvalumab, as well as two PD-1 antibodies, nivolumab, and pembrolizumab, have been approved for various malignancies [311]. The PD-1/PD-1L therapy appears to increase NO release, likely through increased immunological activity. In one study, exhaled NO significantly increased after administration of Nivolumab, suggesting increased lung inflammation, and increases in exhaled NO were especially notable in patients with COPD [190,313]. However, Nivolumab was not associated with an increased risk of COPD exacerbations or spirometry decline [313]. Interestingly, increased expression of PD-1L on cancer cells is thought to be mediated by a HIF-1α mechanism [314,315]. Treatment with a NO donor in combination with a PD-1/PD-1L therapy can potentially reverse this effect of HIF-1α on PD-L1 accumulation within the tumor [314]. Furthermore, the transcription factor YY1 is shown to further mediate the PD-1L expression, which NO can inhibit, further enhancing the effect of immunological therapy [314,316,317]. Inhibition of YY1 by NO may also activate apoptotic pathways, further increasing the effect of immunological therapy and chemotherapy. Preclinical models that target this pathway are promising, as NO modulation combined with PD-L1 therapy shows efficacy against breast cancer and melanoma in vivo [318,319]. However, current ongoing clinical trials (NCT03236935, NCT04095689) are utilizing L-NMMA, a non-selective NOS inhibitor, combined with PD-L1 therapy so the clinical outcomes of PD-1/PD-L1 augmentation with NO are unknown.

Similar to PD1 therapy, other immunotherapies also have critical interactions with NO. One such immunotherapy is anti-Colony Stimulating Factor-1 Receptor (CSF1R). CSF1R responds to its ligands, colony-stimulating factor-1 (CSF1), and interleukin-34 (IL-34) and functions to control macrophage proliferation and survival [320]. CSF1 is one of many macrophage growth factors; however, previous studies found that CSF1R was released during late-stage inflammatory reactions, which the authors suggested was related to the development of M2 macrophages [321]. CSF1R accumulation has been associated with poor patient prognosis [322]. Similarly, TAM M2 anti-inflammatory macrophages have also been associated with cancer progression and poor patient prognosis [323]. Therefore, targeting this pathway with immunotherapy has been suggested to be of great clinical benefit, especially for patients with advanced malignancies [322]. CSF1 application to breast cancer cells showed significant induction of iNOS activity and a corresponding increase in NO [324]. Furthermore, treatment with CSF1 antibodies reversed this effect, suggesting that iNOS was central to the function of the enzyme [324]. Interestingly, this observation clashes with previous studies suggesting that M2 macrophage activation decreases NO-mediated reactive species production [79,325]. The role of NO within this pathway remains poorly understood, and no active clinical trials are ongoing with CSF1R/CSF1 monoclonal antibodies and NO modulators.

## 7. Future Directions

Current therapeutic NO applications are limited by pharmacodynamics, pharmacokinetics, or systemic absorption and toxicity [207,326]. To help overcome these challenges, a number of NO-releasing and -containing compounds have been developed that enhance the pharmacodynamic properties of NO as an anticancer agent. One avenue that has shown efficacy is the conjugation of NO to a number of different molecules, such as non-steriodal anti-inflammatory drugs (NSAIDS) and chemotherapeutics. For example, NO conjugated with NSAIDS and doxorubicin either enhanced cytotoxicity or increased intracellular accumulation of chemotherapeutics within the tumor [207,327,328]. NO hybridization may also be effective in increasing the cytotoxicity of drugs not traditionally regarded as chemotherapeutics, such as lopinavir, an antiretroviral agent [329]. Nevertheless, NO or NO-modifying hybrid drugs may still lack the necessary specificity to engage in tumor-specific targeting and may still be limited by systemic toxicity.

Multiple interesting and innovative mechanisms have been proposed to aid in tumor-specific targeting. One approach is conjugating NO donors and PDE-is to designer antibodies for immunotherapy, which have shown efficacy [330,331,332]. For example, one study designed a NO-releasing antibody against CD24+, a widely expressed hepatocellular cancer marker, and was found to have high cellular uptake and apoptotic activity [332]. Another study validated NO-donating metal complexes conjugated to polyclonal antibodies in vitro and found an 80% increase in cytotoxicity with the NO donor used as an antibody-drug complex (ADC) [331]. Other ADC-utilizing PDE-4is have also shown promise, although the efficacy of novel NO ADCs largely remains unknown, especially when compared to existing treatment regimens [330]. Other approaches utilize specific environmental triggers to encourage NO release from a conjugated compound via an environmental trigger. For example, NO can be conjugated to doxorubicin such that it becomes released when exposed to wavelengths of visible light [333]. Further modification of the NO-drug-releasing compounds to nanoparticles and liposomes has been shown to increase the tumor selectivity, half-life, or wavelength responsiveness of NO-donating drugs combined with macromolecules [207,334]. Other organic and inorganic polymers or porous materials have also been proposed as NO-releasing agents, although their efficacy and biocompatibility are still under study [207,326]. Although preclinical studies of NO-donating macromolecules are promising, few studies have tested the efficacy of these compounds in vivo, suggesting that clinical applications of such compounds are far off. Thus, the future of NO therapy may lie upon clinicians finding the right mechanism to deliver NO alongside chemotherapy.

## 8. Conclusions

The discovery of multiple NO-mediated pathways within cancer has unlocked a number of novel NO-based therapies. Many of these novel therapies center around delivery of NO directly to the tumor and TME. Such localized increases in NO may reverse chemotherapeutic and radiotherapeutic resistance mediated by HIF-1α, although preclinical trials have suggested the efficacy with a number of other mechanisms. However, a primary limitation is the controlled delivery of NO directly to the tumor that tightly regulates localized NO concentration while minimizing side effects. Multiple promising studies have supported the efficacy of NO-releasing biomolecules to aid in NO delivery. Advances in biomaterials, combined with multiple clinical trials demonstrating the efficacy of NO-related therapies alongside radio-, immuno-, and chemotherapy, suggest that the future of NO as an anticancer agent has only begun.

## Figures and Tables

**Figure 1 vaccines-09-00094-f001:**
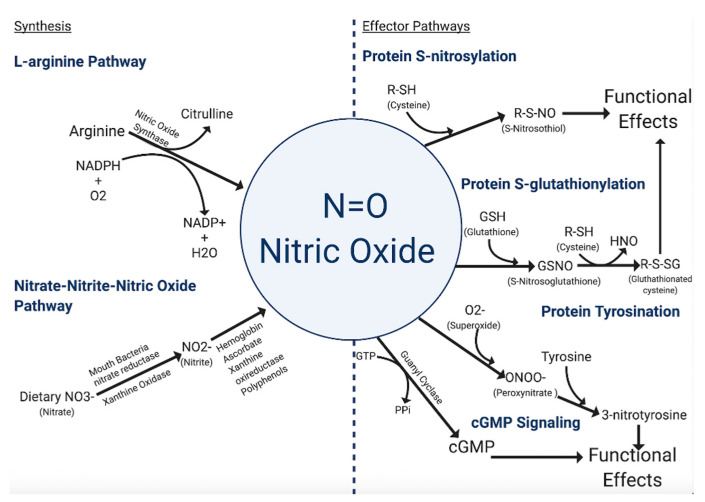
Major synthesis and effector pathways of NO. Synthesis of NO primarily occurs through the action of NOS, although the conversion of dietary nitrogen containing compounds has also been proposed. Once produced, the function of NO can directly upregulate cGMP second messenger signaling pathways and directly modify the function of proteins through nitration. However, in the presence of other ROS such as superoxide, NO may form more reactive intermediates which can further alter the functionality of proteins. The balance of NO with ROS is critical to maintaining proper cellular function and the NO/ROS imbalance is implicated in the pathogenesis of many different diseases.

**Figure 2 vaccines-09-00094-f002:**
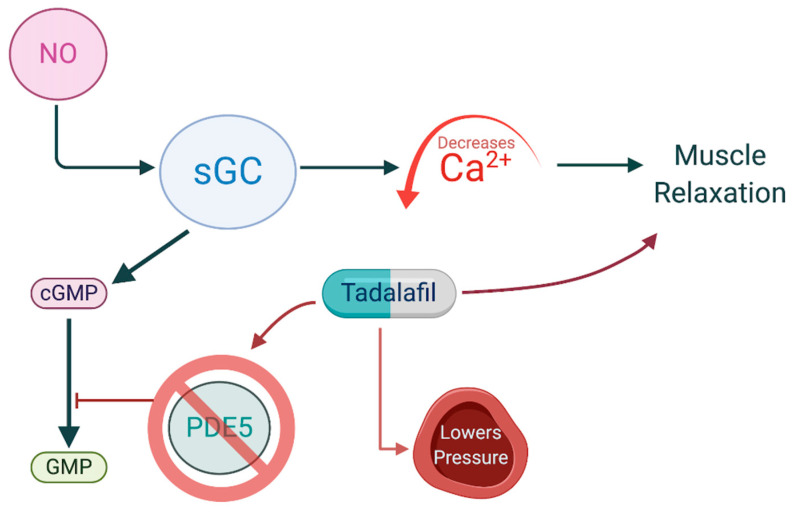
NO and PDE5 inhibitors control cGMP levels, thereby lowering vascular pressure. NO binds Scheme 5. which is responsible for hydrolysis of cGMP. NO, nitric oxide; PDE5, phosphodiesterase-5; cGMP, cyclic guanosine monophosphate.

**Figure 3 vaccines-09-00094-f003:**
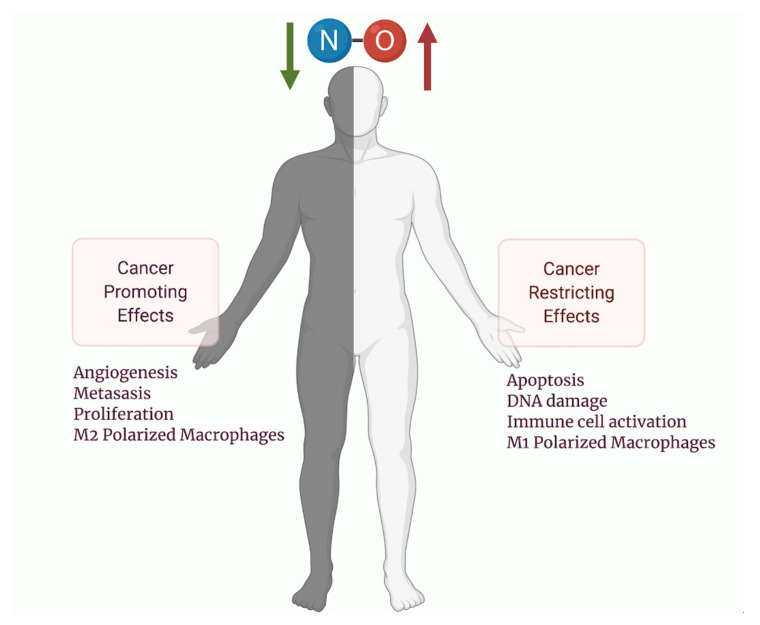
Concentration-dependent effects of NO in cancer. Low NO improves molecular processes that maintain normal physiology but may influence cancer progression of already established cancers, such as proliferation, angiogenesis, metastasis, and switch to immunologically suppressive immune cell types, such as M2 macrophages. High NO influx leads to DNA damage, p53 activation, and nitrosative stress, which may promote carcinogenesis initially, but in already-established cancers, high NO promotes processes that activate immunity and improve chemotherapeutic efficacy. NO, nitric oxide.

**Figure 4 vaccines-09-00094-f004:**
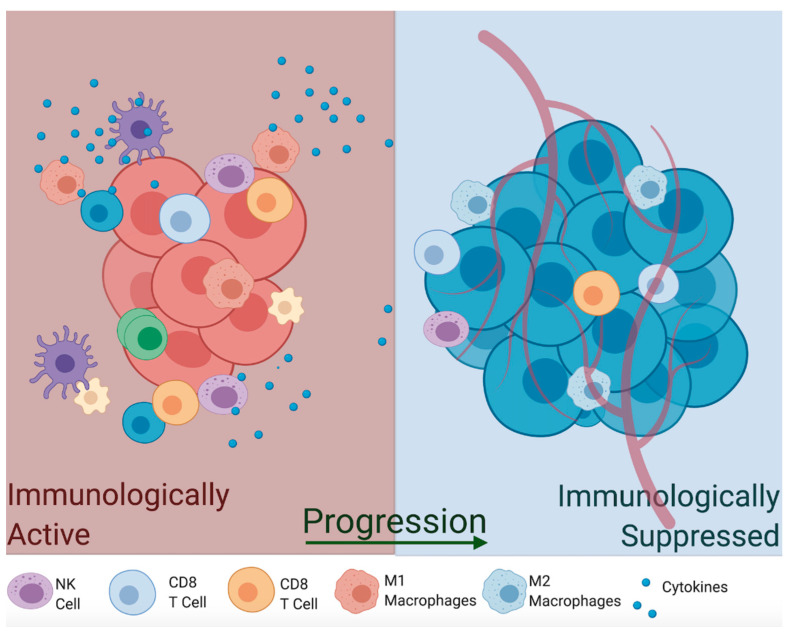
NO promotes inflammatory tumor microenvironment by increasing polarization of M1 macrophages, which in turn produce NO through upregulation of iNOS, and other immune cells that can effecTable 1. macrophages and other pro-inflammatory cell types, to immunologically suppressed tumors that favor M2 macrophage switch which in turn downregulate iNOS production and promote immunosuppression, angiogenesis and are resistant to immunotherapy. TNFα, tumor necrosis factor-alpha; INFγ, interferon-gamma; IL-1β, Interleukin-1b.

**Figure 5 vaccines-09-00094-f005:**
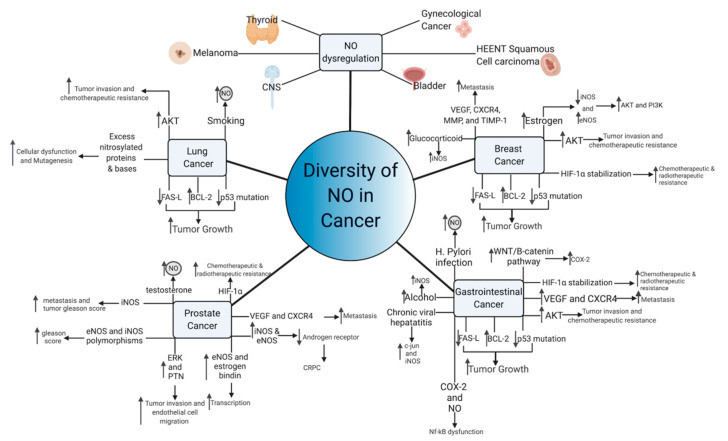
Diversity of NO functioning in cancer. Multiple NO-mediated cancer pathways contribute to cancer growth and metastasis. Common pathways of NO-mediated mutagenesis and cancer growth, include p53 mutation, AKT upregulation, and VEGF induction, among others. Aberrant NO signaling also occurs in response to common carcinogens, including viruses, alcohol, and tobacco, suggesting that NO may lie in a common carcinogenic pathway shared by these compounds. In hormone-sensitive tumors, NO paradoxically functions to transmit the hormonal growth signals and can make the tumor hormone-insensitive.

**Table 1 vaccines-09-00094-t001:** Major classes of NO donors and their examples of their proposed clinical uses.

Class	Mechanism of Action [207,208]	Examples [207]	Known or Potential Clinical Uses
Endogenous NO precursors	Provide excess reagents or intermediates for NO synthesis. Direct antioxidant. Reversal of NOS inhibition.	L-Arginine, ω-hydroxy-L-arginine (NHA)	Stroke [209], Pulmonary hypetesion [210]
Organic Nitrates	Stimulation of guanylyl cyclase pathway, through unknown intermediate. May activate enzymatic NO production through cytochrome P450 or glutathione-*S*-transferase. Non-enzymatic thiol activity also described, although reaction proceeds at slower rate	Glyceryl trinitrate (GTN), isosorbide mononitrate (ISMN), pentaerythritol tetranitrate (PETN). nicorandil	Cardiac Angina [211], acute coronary syndrome [212], heart failure [213]
Organic Nitrites	Activation of NO signaling pathways	Butyl nitrite (BN), isobutyl nitrite (ISBN), *tert*-butyl nitrite (TBN), amyl nitrite (AMN), isoamyl nitrite (IAMN)	Minimal clinical uses due to cardiotoxicity and cytotoxicity
Metal Complexes	Direct release of NO. NO bound to metals such as iron is prone to nucleophilic attack	Sodium Nitroprusside (SNP)	Hypertensive Emergency, cardiac and aortic surgery [214,215], heart failure [216], acute coronary syndrome [217], pheochromocytoma [218]
Diazeniumdiolates	Spontaneously decompress to 2 molecules of NO	Methylamine hexa-methylene methylamine NONOate (MAHMA/NO), diethylamine NONOate (DEA/NO), proli NONOate (PROLI/NO) and diethylenetriamine NONOate (DETA/NO)	No current clinical uses
Sydnonimines	Spontaneous enzymatic degradation to O2 and NO. Also known as ONOO-donors. cGMP independent activation. Increase in K+ channel activity	3-morpholinosydnonimine (SIN-1), molsidomine (N-ethoxycarbonyl-3-morpholinosydnonimine)	Minimal clinical trial evidence to support use [207]
S-nitrosothiols	Cleavage of S-NO bond releases NO and disulfide. S-nitrosation of cellular components. cGMP stimulation.	S-nitrosoglutathione (GSNO), N-acetylpenicill-amine (SNAP), trityl S-nitrosothiol, (Ph3SNO) and tert-butyl S-nitrosothiol (tButSNO)	Onychomycosis [207], sexual dysfunction [219], antiplatelet agent [220], cardiac surgery [221], cystic fibrosis [222]

**Table 2 vaccines-09-00094-t002:** Clinical Studies involving NO donors and Cancer.

Completed Clinical Studies					
NO Drug	Cancer Type	Combination Treatment	Compared with	Effect	Reference
Transdermal Nitroglycerin	Stage IV NSCLC	Paclitaxel, carboplatin, and bevacizumab	Combination treatment minus nitroglycerin	Increased response rate in the nitroglycerin group. Increased overall survival and progression-free survival in nitroglycerin group that did not reach statistical significance	[232]
Transdermal Nitroglycerin	Stage III/IV NSCLC	Vinorelbine and cisplatin	Combination treatment minus nitroglycerin	30% higher response rate in those treated with transdermal nitroglycerin and longer time to disease progression	[226]
Transdermal Nitroglycerin	Stage IIIB/IV NSCLC	Vinorelbine and cisplatin	Combination treatment minus nitroglycerin	Higher overall response rate and disease control rate in nitroglycerin arm. Time to progression and overall survival were similar	[233]
Transdermal Nitroglycerin	Stages IIIA and IIIB NSCLC	Vinorelbine and cisplatin and radiotherapy	None: Non-randomized	75% overall response rate after chemotherapy and radiotherapy. Median progression-free survival of 13.5 months (95% CI, 8.8–18.2), while the median overall survival was 26.9 months	[230]
Transdermal Nitroglycerin	PSA recurrent prostate cancer after definitive radiotherapy or radical prostatectomy	N/A	N/A	The mean PSA doubling time of the entire cohort increased to 31.8 months from 13.2 months before starting treatment	[228]
Transdermal Nitroglycerin	Operable clinical stage T3-4, or T1-4 node-positive, M0 rectal adenocarcinoma	5-fluorouracil and radiation therapy prior to surgery	None: Phase I Trial	Pathological Complete response of 17%. Only one patient experienced side effects attributed to nitroglycerin	[229]
IV Nitroglycerin	Barcelona clinic liver cancer stage A/B hepatocellular carcinoma	Doxorubicin emulsified in Lipiodol followed by Transcatheter arterial embolization (TAE) and transcatheter arterial chemoembolization (TACE)	Combination treatment minus nitroglycerin	Greater change in lesion size from baseline was observed for nitroglycerin group. Nitroglycerin therapy group showed higher concentration of lipiodol (and thus chemotherapeutic) inside the lesion.	[227]
Isosorbide mononitrate	Stage IIIB/IV NSCLC	Irinotecan plus cisplatin and Irinotecan plus capecitabine	Combination treatment minus Isosorbide mononitrate	Isosorbide mononitrate addition did not improve outcomes to either treatment group	[234]
Isosorbide mononitrate	T1-T4 Oral Squamous Cell Carcinoma	Surgery after drug administration	Surgery alone	No difference in Ki-67 tumor staining was noted	[235]
**Ongoing Clinical Studies without Posted Results**					
**Drug**	**Cancer Type**	**Combination Treatment**	**Compared with**	**Completed**	**Clinical Trial Number**
Transdermal Nitroglycerin	NSCLC with brain metastases	Whole brain radiation	Combination treatment minus nitroglycerin	Yes	NCT04338867
IV Nitroglycerin	Pediatric Retinoblastoma	Intra-arterial chemotherapy	Normal Saline and combination treatment	No	NCT04564521
Dietary Arginine	Colon Cancer	Nutritional Supplement Prior to Surgery	Surgery alone	Yes	NCT04564521
Dietary Arginine	Stage III/IV Head and Neck Cancer	omega-3 fatty acids and nucleotides oral supplement with cisplatin and radiation therapy	Cisplatin and radiation alone	Yes	NCT00559156
Dietary Arginine	Bladder Cancer	Radical cystoscopy	Radical cystoscopy alone	Yes	NCT02655081
Dietary Arginine	unresectable metastatic brain tumors	Radiation	Radiation alone	Yes	NCT02844387
Nicorandil	stage II—IV NSCLC	Radiation	Radiation alone	Unknown	NCT02809456

N/A: No additional treatment.

**Table 3 vaccines-09-00094-t003:** In vivo and in vitro preclinical studies involving NO donors.

In Vivo Preclinical Studies				
NO Drug	Cancer Type	Additional Treatment	Effect	Reference
Isosorbide mononitrate (ISMN) and isosorbide dinitrate (ISDN)	Lewis Lungcarcinoma in mice	None	Inhibited angiogenesis and tumor growth	[241]
Nicorandil	Sprague-Dawley rats without malignancy	Bleomycin	Reduced lung inflammation and fibrosis	[242]
DETA/NO	BALB/c female mice with mammary adenocarcinoma in a experimental metastasis model	CORM-A1	Inhibited the EMT	[243]
PROLI/NO	Sprague–Dawley rats with Brain Gliomas	Carboplatin	Increased blood brain barrier permeability and survival	[239]
S-nitrosoglutathione	Mouse xenograft model with head and neck squamous cell carcinoma	Radiation and cisplatin	Decreased tumor growth and enhanced therapySTAT3 inhibition	[244]
S-nitrosoglutathione	C57BL/6J mice with castration resistant prostate cancer	None	Decreased tumor burdenIncreased the expression of LH, FSH, and testosteroneDecreased M2 TAM and increased T1 TAMAndrogen receptor downregulation	[182]
**In Vitro Preclinical Studies**				
**NO Drug**	**Cancer Type**	**Additional Treatment**	**Effect**	**Reference**
ω-hydroxy-L-arginine	MDA-MB-468 Breast Cancer Cells	None	Decreased cellular proliferative and Increased apoptosis	[245]
isosorbide mononitrate	HCT116 and SW620 colon cancer cells	Aspirin	Synergistic effect of therapy on inhibition of cell growth	[246]
Sodium Nitroprusside/L-arginine	AGS gastric cancer cell line	None	Inhibition of Epidermal growth factorActivation of type II cGMP-dependent protein kinase	[247]
Sodium Nitroprusside	SGC-7901, AGS, MKN45 and MKN28 gastric cancer cells	None	Increased apoptosis through TRAIL cytotoxicity	[248]
Sodium Nitroprusside	TSCCa tongue oral squamous cell carcinoma	None	Concentration-dependent cytotoxicity and increased apoptosis	[249]
Sodium Nitroprusside	N1E-115 neuroblastoma cells	Cycloheximide	Induction of cell death	[250]
Sodium Nitroprusside	HeLa cervical cancer cells	GS28 siRNA (siGS28) transfection	Inhibited cytotoxic responseIncreased ERK	[251]
Sodium Nitroprusside	HepG2 and Hep3B Hepatocellular carcinoma cells	Deferoxamine	Induced apoptosisApoptoic effect inhibited by deferoxamine pretreatment	[252]
Sodium Nitroprusside	Eight human pancreatic tumour cell lines	Radiation	Increased sensitivity to radiotherapy	[253]
Sodium Nitroprusside	SK-MEL-28 and WM793 Melanoma Cell Lines	Arginine Deprivation	Increased therapeutic effect	[254]
Sodium Nitroprusside	SH-SY5Y neuroblastoma cells	2-day light-exhausted compound SNP(ex)	Increased apoptosisIncrease in p53 activation of both SNP and SNP(ex)	[255]
S-nitroso-N-acetylpenicillamine and sodium nitroprusside	CHP212 neuroblastoma cells	Deferoxamine	Induced apoptosis, although with a different time to inhibition.	[256]
ruthenium nitrosyl complex trans-[Ru(NO)(NH3)4(py)](PF6)3]	HepG2 Liver cancer cells	None	Induced apoptosis	[239]
Spermine nitric oxide complex hydrate (SPER/NO)/ diethylenetriamine nitric oxide adduct (DETA/NO)	SK-OV-3 and OVCAR-3 ovarian cancer cell	None	Enhanced cytotoxicity and inhibited apoptosisDownregulation of STAT3 and AKT	[249]
DETA-NONOate	MDA-MB-231 breast cancer cells	None	Induced G1 phase growth arrestDownregulation of cyclin D1Hyperphosphorylation of RB	[257]
DETA-NONOate	MDA-MB-468 breast cancer cells	Farnesyltransferase inhibitor	Induced apoptosisCytochrome-c release and caspase 3/9 activation	[236]
DETA-NONOate	MDA-MB-231, MDA-MB-157, MDA-MB-436, HCC-1806, HCC-70, MDA-MB-468, HCC-1395 and BT-549 breast cancer cell	None	Increased mitochondrial induced apoptosis in African American cancer cell lines but not caucasian cells lines	[258]
DETANONOate	DU145 and PC-3 Prostate cancer cells	None	Prevented the EMTSNAIL/EMK inhibition	[240]
SNAP and DETA-NONOate	Myeloid derived suppressor cells	None	Inhibited cancer antigen presentation to CD4+ T cells	[259]
Sodium nitroprusside, S-nitroso-N-acetylpenicilamine, S-nitrosoglutathione, (+/−)-(E)-methyl-2-[(E)-hydroxyimino]-5-nitro-6-methoxy-3-hexeneamide, and iNOS transfection	PC-3MM2, LNCaP, and DU145 prostate cancer cells	NO scavengers	Inhibited TGF-B productionSodium nitroprusside and iNOS transfection were ablated with the NO scavengers	[260]
S-nitrosoglutathione	HCT116 and SW620 colon cancer cells	None	Increased apoptosisActivation of ERK1/2 and p38 kinase	[261]
S-nitrosoglutathione	MIAPaCa-2, HCT-116, Panc-1, MCF-7, HT-29 cell lines and AGS cells	U0126 MEK Inhibitor	Growth inhibition through EGFR, IGF-1, and AKT signallingGrowth inhibition increased with U0126 MEK inhibitor	[262]
S-nitrosoglutathione	A549 and NCI-H1299 lung cancer cells	None	Growth inhibition via Prdx2 and AMPK	[263]

**Table 4 vaccines-09-00094-t004:** Clinical Trials Involving PDE-is. Studies that tested the efficacy of PDE-is at preventing side effects of therapy (i.e., ED after radical prostatectomy) are not included.

Completed Clinical Studies					
Phosphodiesterase Inhibitor	Cancer Type	Combination Treatment	Compared with	Effect	Reference
Tadalafil	T1-T4 Oral Squamous Cell Carcinoma	Surgery	Surgery Alone	Myeloid-derived suppressor cells and regulatory T cells were reduced in the blood and tumor, although effect was maximized at the intermediate dose	[281].
Tadalafil	Primary or Secondary Stage III or IV Head and Neck Squamous Cell Carcinoma	Anti-MUC1 Vaccine/ Anti-Influenza Vaccine	Vaccine Alone/Tadalafil Alone	There were no significant adverse effects of combination therapy. Immunohistochemical analysis shows decreased immune cell exclusion from inside the tumor	NCT02544880 (Active, non-recruiting) [282]
Tadalafil	Invasive head and neck squamous cell carcinoma	None	Placebo	An increase of ex vivo T cell expression and reduction in myeloid-derived suppressor cells were noted, suggesting reversal of tumor mediated immune suppression	[283]
Tadalafil	Invasive head and neck squamous cell carcinoma	None	Placebo	One patient in the tadalafil arm died, although no other adverse events were reported in either treatment group	NCT01697800
Exisulind (PDE-2/5i)	Stage IIIB/IV or recurrent non-squamous cell lung cancer	Carboplatin and Gemcitabine	None	Median progression-free survival was 4.7 months while median overall survival was 9.0 months. Combination therapy was well tolerated with the goal of the primary endpoint being met	[284]
Exisulind (PDE-2/5i)	Metastatic breast cancer	Capecitabine	Capecitabine alone	In pretreated patients with other therapies, the addition of exisulind is similar to that of capecitabine alone	[285]
Exisulind (PDE-2/5i)	Solid Malignancy	Docetaxel	None	Combination therapy was well tolerated	[285,286]
Exisulind (PDE-2/5i)	Castration-resistant prostate cancer	Docetaxel	None	Overall survival and progression-free survival were similar to other studies of chemotherapy alone	[287]
Exisulind (PDE-2/5i)	Metastatic castration-resistant prostate cancer	Docetaxel	None	Low likelihood of benefit of exisulind to docetaxel therapy	[288]
CP-461 (PDE-2/5i)	Advanced Solid Malignancy	Usual Treatment	None	Four of 21 patients displayed stable disease and CP461 was well tolerated	[288,289]
Roflumilast (PDE-4i)	Advanced B-cell malignancy	prednisone	None	PI3K activity was suppressed in over 75% of patients, with 66% exhibited partial response or disease stability	[290]
Theophylline (non-selective PDE-i)	Metastatic castration-resistant prostate cancer	Abiraterone/prednisone	Combination Therapy with Dextromethorphan	One subject in the theophylline group experienced a grade 3 increase in alkaline phosphatase	NCT01017939
RA-233( Dipyridamole derivative)	Non-small cell lung carcinoma, small cell lung carcinoma, extensive colon adenocarcinoma	Multiple	Combination Therapy Alone	RA-233 significantly extended median survival in only this with NSCLC limited to 1 hemithorax	[291]
Dipyridamole ( platelet PDE-5/6i	stage II or III unresectable adenocarcinoma of the pancreas	5-Fluorouracil, leucovorin, mitomycin C	None	Median Survival of 13.8 months and an overall response rate of 26%. Six patients underwent curative resection, 2 of whom did not experience disease recurrence	[292]
Aminophylline/ theophylline	B Cell Chronic Lymphocytic Leukemia	None	None	Dose-dependent and time-dependent apoptosis was noted in 45% of patients, who experienced a longer progression-free survival time	[293]
**Ongoing Clinical Studies without Posted Results**					
**Drug**	**Cancer Type**	**Combination Treatment**	**Compared with**	**Completed**	**Clinical Trial Number**
Sildenafil	IIIB or IV non-small cell lung cancer	paclitaxel/carboplatin	Combination Therapy Alone	Yes	NCT00752115
Sildenafil	Advanced solid tumors	Regorafenib	None	Yes	NCT02466802
Sildenafil	Kidney Cancer	Surgery after treatment	Placebo and Surgery	Yes	NCT01950923
Sildenafil	Waldenstrom’s Macroglobulinemia	None	None	Yes	NCT00165295
Sildenafil	WHO Grade III or IV Brain Glioma	Valproic Acid/sorafenib	None	Active-Not Recruiting	NCT01817751
Tadalafil	Refractory hepatocellular carcinoma and pancreatic/colorectal cancer with liver metastasis	Oral vancomycin/nivolumab	None	Recruiting	NCT03785210
Tadalafil	Recurrent or Metastatic Head and Neck Cancer	Pembrolizumab	None	Recruiting	NCT03993353
Tadalafil	Resectable Head and Neck Cancer	Nivolumab/surgery	Combination Therapy Alone	Active-Not Recruiting	NCT03238365
Aminophylline (non-selective PDE-i)	Bladder Cancer	BCG Vaccine	None	Yes	NCT01240824
pentoxifylline (non-selective PDE-i)	grade IV astrocytoma/glioblastoma multiforme	hydroxyurea/radiotherapy	Combination Therapy Alone	Yes	NCT00019058
Dipyridamole	Stage III/IV ovarian carcinoma refractory to platinum chemotherapy	Intraperitoneal methotrexate	Unknown	Unknown	NCT00002487
Thymoquinone (PDE-1i)	Premalignant oral Lesions	None	Placebo	Yes	NCT03208790
Vesnarinone (PDE-3i)	Kaposi Sarcoma	None	Unknown	Yes	NCT00002131
PBF-999 (PDE-10i)	Advanced metastatic solid tumor	Usual Treatment	None	Recruiting	NCT03786484

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
