# Peer review of "Current Advances of Nitric Oxide in Cancer and Anticancer Therapeutics"

_vaccines, 2021, doi:10.3390/vaccines9020094_

Round 1

Reviewer 1 Report

The review is very well organized and nicely framed out. I appreciate that authors have put substantive effort in chalking it out. Few points have to be revised :

Authors have not mentioned much about the chemistry of NO. More details on synthetic procedures is necessary, providing a scheme explaining different synthetic routes would be helpful. Similarly, more detailed information on variety of post translational modifications of NO along with schematic presentation will clarify the picture.

A paragraph as well as a diagram should be added highlighting special features and chemical properties of NO, for which it is useful in anticancer therapy.

Section 5 describes role of NO for different type of cancers. Illustrating each section with a figure will help in better understanding of the matter.

Reviewer 2 Report

This is a thorough and  detailed manuscript of NO biology in carcinogenesis and the kinetics within  the tumor microenvironment. Anticancer therapies influencing NO levels are described and clinical trials are summarized. Trials without results are shown and may serve to provide an overall idea to readers.

I have two minor concerns.

Section 5.0 "NO donating drugs may overwhelm the physiological mechanisms underlying tumorigenesis and actually serve to inhibit cancer growth" . The authors are  suggesting  a  generalized and  vague description of  'overwhelm' which is inaccurate considering that specific  signaling and cellular mechanisms such as  apoptosis have been reported.  

In Fig 3 legend text where " cold tumors that favor M2 macrophage switch which in turn downregulate iNOS production .. " there is diagram of a balance or scale at the top middle representing NO levels. Related to this text,  it is not clear if the scale shown in the figure that is shown weighed down on the right hand side should be depicted  differently, if it is a indeed meant to display lower levels of  iNOS/NO. 

Is fig 2, the word Relaxed is not clear.  Both figures should have similar fonts.

Reviewer 3 Report

Joel et al. have extensively covered and compiled an inclusive literature survey that is related to NO, the role of NO in anti-cancer, and advanced therapeutics. Here few additions and minor comments. PLease include them.

1) Minor: Please check the ref. 210 (capital letters) for the font. Please follow the journal's instructions closely for reference and other instructions for consistency.

2) Please draw a conclusive figure for section 5 (NO in different cancer type) you can draw a circle diagram or tree diagram i.e. NO in cancer, then different cancer type example and in each one how NO affects or regulate important pathways or if NO can be diagnostic target (A summarized figure will be easy to a basic reader to understand the versatility of NO in cancer therapy/diagnosis/theranostic)

3) section: NO and Angiogenesis in the TME Please find and add a recent article published in clinical cancer research based on NO/PDE5i/prostate cancer. You must include it is an interesting study (PubMed 32847934). 

4) Table 3: column-Effect: Please try to concise the write-up. Short, concise, and summarized wording will be suitable to the reader. Looking a little wordy, so use some abbreviations to cut-short space.   

5) Please try to review if any antibody-drug conjugates are included for targeting NO. 

6) Figure 1 can be remodified by showing some/few names of important and well-known pathways that mainly involves in both cancer-promoting and cancer restricting site (if possible)     
